# Development and Analysis of Key Components of a Multi Motion Mode Soft-Bodied Pipe Robot

Ning Wang [ID], Yu Zhang *, Guofeng Zhang, Wenchuan Zhao and Linghui Peng

School of Mechanical Engineering, Shenyang University of Technology, Shenyang 110870, China;
sut_wangning@126.com (N.W.); xuehouniao1981@126.com (G.Z.); zhao_wenchuan@126.com (W.Z.);
penglinghui0130@163.com (L.P.)
* Correspondence: zhangyu@sut.edu.cn

**Abstract:** In order to enhance the environmental adaptability of peristaltic soft-bodied pipe robots, based on the nonlinear and hyperelastic characteristics of silicone rubber combined with the biological structure and motion characteristics of worms, a hexagonal prism soft-bodied bionic actuator is proposed. The actuator adopts different inflation patterns to produce different deformations, so that the soft-bodied robot can realize different motion modes in the pipeline. Based on the Yeoh binomial parameter silicone rubber constitutive model, the deformation analysis model of the hexagonal prism soft-bodied bionic actuator is established, and the numerical simulation algorithm is used to ensure both that the drive structure and deformation mode are reasonable, and that the deformation analysis theoretical model is accurate. The motion and dynamic characteristics of the prepared hexagonal prism soft-bodied bionic actuator are tested and analyzed, the motion and dynamic characteristic curves of the actuator are obtained, and the empirical deformation formula of the actuator is fitted. The experimental results are consistent with the deformation analysis model and numerical simulation result, which shows that the deformation analysis model and numerical simulation method are accurate and can provide design methods and reference basis for the development of a pneumatic soft-bodied body bionic actuator. The above research results can also prove that the hexagonal prism soft-bodied bionic actuator is reasonable and feasible.

**Keywords:** silicone rubber; pneumatic; pipeline robot; soft-bodied bionic actuator; soft-bodied robot

## 1. Introduction

With the continuous development of society, pipeline transportation has been widely used in the fields of petroleum, chemical industry, energy, food processing, urban water supply and drainage, agricultural irrigation, nuclear industry, etc. However, due to the influence of chemical corrosion of transmission mediums, force majeure natural disasters, and their defects, serious accidents may occur, such as environmental pollution, flammable explosions, energy waste, and so on [1]. It is necessary to inspect, maintain and clean the inside of the pipeline regularly. Due to the complexity and large number of pipe networks, the traditional pipeline inspection workload is large and efficiency is low, and some pipeline locations cannot be reached to perform inspection. Therefore, pipeline robots are required to perform pipeline inspection and maintenance tasks.

A pipeline robot is a mechanical, electrical and instrument integrated system that can move along the inside or outside of small pipelines, carry one or more sensors and operating machinery, and carry out a series of pipeline operations under the remote control of workers or computer automatic control [2]. According to different motion forms, they can be divided into mobile pipeline robots [3,4], wheeled pipeline robots [5–7], tracked pipeline robots [8,9], supported pipeline robots [10], walking pipeline robots [11–13], peristaltic pipeline robots [14–16] and spiral pipeline robots [17,18]. Traditional pipeline robots are generally made of rigid materials. Due to the poor adaptability of a rigidly structured body

to the environment, it is difficult for the robot to work in the rugged and complex pipeline environment, and the rigid materials are in contact with the pipeline, which could easily cause damage to the pipeline contact surface and aggravate the damage to the pipeline [19].

The emergence of soft-bodied robots provides a new idea to solve these problems. The key of a soft-bodied robot lies in its different driving methods [20,21], such as pneumatic driving, shape memory alloy driving and dielectric elastomer driving. Among them, pneumatic is more popular because it has the advantages of a simple driving mode, high driving efficiency and strong pressure resistance. In recent years, scientists have designed a series of peristaltic soft-bodied pipe robots driven by air pressure by studying the motion mechanism of worms in nature and combining with soft-bodied robot technology.

For example, the Calderón team from the University of Southern California designed an earthworm-like soft-bodied body pipe peristaltic robot using two contraction actuators and one extension actuator [22]; Connolly et al. from Harvard University designed a segmented worm-like soft-bodied pipe robot composed of compression actuator, extension actuator, compression actuator and torsion actuator [23]; Yamazaki et al. from the Central University of Japan designed an earthworm type 25A pipeline inspection robot using pneumatic actuator [24]; Mohit S. Verma of Harvard University developed a pneumatic soft pipe robot using buckling pneumatic actuators (vacuum-actuated muscle-inspired pneumatic structures, or VAMPs), with which the tube climber can navigate through a tube with turns, inclines, and varying diameters [25]; Philip Wai Yan Chiu et al. from the University of Hong Kong proposed a pneumatic soft-bodied robot for gastrointestinal examination, which can move peristaltically in a tubular environment such as a colon [26]; Wang Xueqian et al. from Tsinghua University developed a pipe crawling robot with parallel soft-bodied body actuators [27]; Xi Zuoyan et al. from Harbin Institute of Technology designed a soft-bodied robot that can crawl in the pipeline by using the series combination of bending actuator and telescopic actuator [28]; Jiang Cheng et al. from Donghua University designed a worm-like soft-bodied pipe robot based on fabric paper composites [29]. The above-mentioned peristaltic soft-bodied pipe robots often have a single motion mode and an insufficient ability to deal with environmental changes, and the research theory and method cannot be applied universally.

Our laboratory has developed a multi-motion mode peristaltic soft-bodied pipe robot [30], which is composed of a hexagonal prism soft-bodied bionic actuator, elongation soft-bodied actuator, and hexagonal prism soft-bodied bionic actuator in series. By adopting different inflation patterns for the hexagonal prism soft-bodied bionic actuator, the robot can realize different motion modes in pipes of different diameters. The key component to realizing the multi-motion mode of the peristaltic soft-bodied pipe robot is the hexagonal prism soft-bodied bionic actuator. In this paper, structural design, mechanical modeling, numerical simulation algorithm verification, physical model preparation, and experimental test analysis are carried out. Finally, it is determined that the research method, process, and results of the hexagonal prism soft-bodied bionic actuator are reasonable and feasible, which lays a foundation for the further study of peristaltic soft-bodied pipe robots.

## 2. Structural Design and Deformation Analysis of Hexagonal Prism Soft-Bodied Bionic Actuator

### 2.1. Principal Design

The movement diagram of the peristaltic soft-bodied pipe robot is shown in Figure 1, and the movement direction of the robot is to the right. By adopting different inflation patterns for the hexagonal prism soft-bodied bionic actuator, radial expansion deformation or axial bending deformation can be produced respectively: when facing a pipe with a running pipe diameter similar to the size of the hexagonal prism soft-bodied bionic actuator, the actuator can adopt the radial expansion deformation mode, and the robot can move in the pipe in a manner similar to earthworm peristalsis (Figure 1a); when the size of the pipe is much larger than that of the hexagonal prism soft-bodied bionic actuator, the actuator can

adopt the axial bending deformation mode, and the robot adopts the flexion and extension motion similar to inchworm larva in the pipe (Figure 1b).

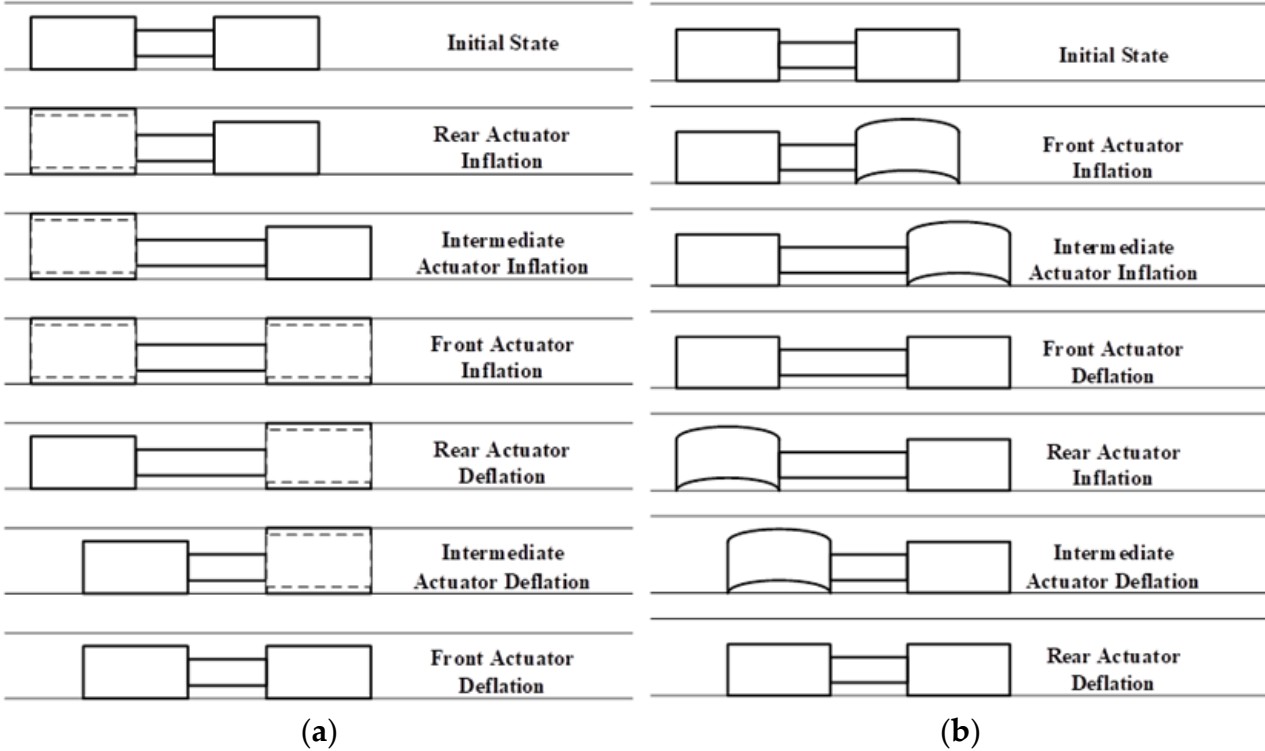

(**a**)                  (**b**)

**Figure 1.** Motion diagram of peristaltic soft-bodied pipe robot. (**a**) Radial expansion motion. (**b**) Axial bending motion.

The structure of the hexagonal prism soft-bodied bionic actuator is shown in Figure 2; the main components are six basic rectangular cavities with the same structural size and one non-retractable regular hexagonal silicone rubber column. The rectangular bottom surfaces of the six rectangular cavities are respectively bonded with the six rectangular sides of the regular hexagonal silicone rubber column through silicone rubber adhesion technology. Each rectangular cavity is composed of 11 air cavity structures. In addition, in order to enhance the movement ability and reduce the weight of the actuator, and consider the integration of power source and control system, the regular hexagonal silicone rubber column adopts a circular hollow structure.

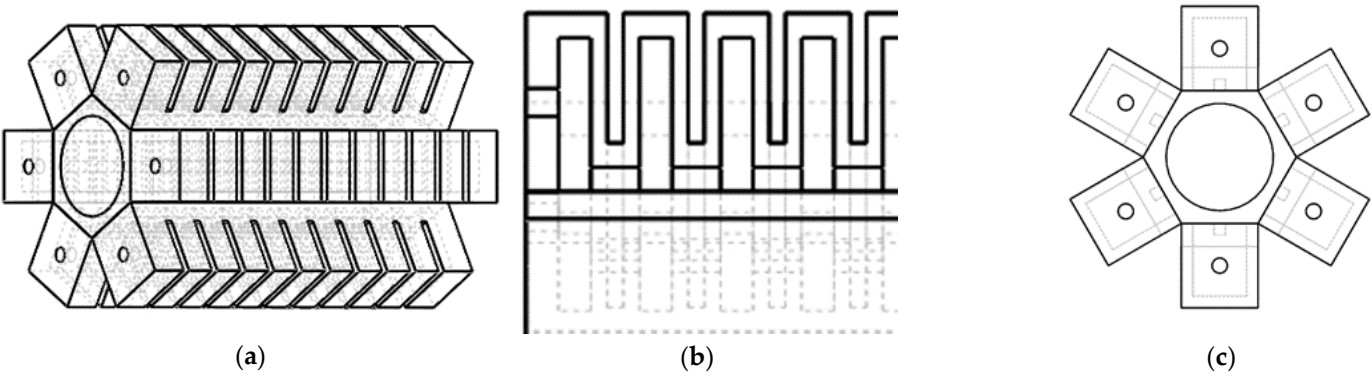

(**a**)                  (**b**)                  (**c**)

**Figure 2.** Structure diagram of hexagonal prism soft-bodied bionic actuator. (**a**) Overall structure of actuator. (**b**) Partial view of actuator section. (**c**) Actuator end view.

The main structural parameters of the hexagonal prism soft-bodied bionic actuator are shown in Table 1. The design requires that the wall thickness of the expansion wall of the air cavity structure is less than the other wall thickness, so as to ensure that the deformation of the actuator is mainly realized by the interaction between the expansion walls.

**Table 1.** Main structural parameters of the actuator.

| No. | Size | Num |
|-----|------|-----|
| 1 | Height of rectangular cavity $h_2$ | 19 mm |
| 2 | Length of rectangular cavity $w_2$ | 14 mm |
| 3 | Width of rectangular cavity $l_2$ | 4 mm |
| 4 | Thickness of expansion wall of rectangular cavity $l_3$ | 2 mm |
| 5 | Side length of hexagonal silicone rubber column $a$ | 20 mm |
| 6 | Length of hexagonal silicone rubber column $l$ | 112 mm |
| 7 | Dimensions of hollow structure of hexagonal silicone rubber column $r$ | 14 mm |
| 8 | Number of air chamber structures $N$ | 11 |

### 2.2. Deformation Mode Analysis

The hexagonal prism soft-bodied bionic actuator can produce axial bending deformation and radial expansion deformation by different inflation patterns. The principle is as follows: when six rectangular cavities are filled with gases at different pressures, the deformation degree of each rectangular cavity is different, and the driver will bend towards the side with a small deformation degree of the rectangular cavity, allowing the soft-bodied actuator to realize axial bending deformation; when the same air pressure is applied to all six rectangular cavities at the same time, the deformation degree of each cavity is the same, the actuator will not bend, but will increase the overall radial size and realize radial expansion and deformation. According to the requirement of ensuring the stable movement of the robot in the pipeline in reference [30], this paper only analyzes the bending deformation of the hexagonal prism soft-bodied bionic actuator with two cavities inflated. The inflation deformation diagram of hexagonal prism soft-bodied bionic actuator is shown in Figure 3.

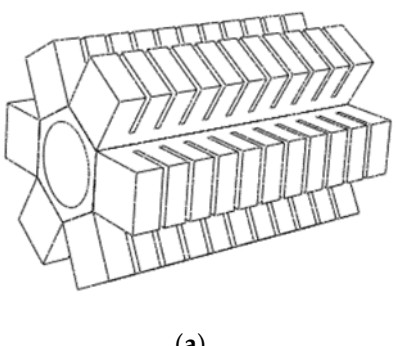

(**a**)

**Figure 3.** *Cont.*

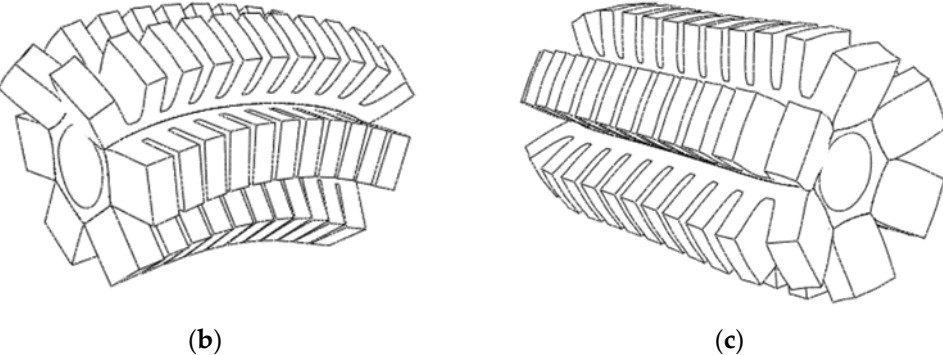

$$(\mathbf{b}) \qquad\qquad (\mathbf{c})$$

**Figure 3.** Deformation diagram of hexagonal prism soft-bodied bionic actuator. (**a**) Status of actuator not inflation. (**b**) State of bending deformation. (**c**) State of expansion deformation.

### 3. Mechanical Model Analysis

*3.1. Constitutive Model of Silicone Rubber*

The soft-bodied actuator is made of silicone, which is a material with non-linear, superelastic, and large deformation, and shows almost no change in volume (that is, it is incompressible). Its mechanical properties are usually expressed and analyzed by the strain energy density function. The most commonly used strain potential energy models include the Mooney–Rivlin model, Yeoh model, and Ogden model [31–36]. According to the literature, the Yeoh model can fit other different deformation modes according to the data of the uniaxial tensile test, such as uniaxial compression and shear, and is suitable for simulating large deformation. It is the preferred constitutive model for analyzing the deformation of silicone rubber.

For isotropic silicone rubber materials, based on the stress–strain relationship, the constitutive relationship of silicone rubber materials can be expressed by the strain energy density function:

$$
\begin{cases}
W = W(I_1, I_2, I_3) \\
I_1 = \lambda_1^2 + \lambda_2^2 + \lambda_3^2 \\
I_2 = \lambda_1^2\lambda_2^2 + \lambda_2^2\lambda_3^2 + \lambda_1^2\lambda_3^2 \\
I_3 = \lambda_1^2\lambda_2^2\lambda_3^2
\end{cases}
\tag{1}
$$

where $I_1$, $I_2$ and $I_3$ are deformation tensor invariants, and $\lambda_1$, $\lambda_2$ and $\lambda_3$ are the main elongation ratios. For incompressible silicone rubber materials : $I_3 = \lambda_1^2\lambda_2^2\lambda_3^2 = 1$ .

Assuming that the material does not deform in the width direction, then $\lambda_3 = 1$, we can get:

$$
\lambda_1 = \frac{1}{\lambda_2}
\tag{2}
$$

$$
I_1 = I_2 = \lambda_1^2 + \frac{1}{\lambda_1^2} + 1
\tag{3}
$$

The strain energy density function of the typical second-order Yeoh constitutive model is as follows:

$$
W = C_{10}(I_1 - 3) + C_{20}(I_1 - 3)^2 = C_{10}\left(\lambda_1 - \frac{1}{\lambda_1}\right)^2 + C_{20}\left(\lambda_1 - \frac{1}{\lambda_1}\right)^4
\tag{4}
$$

According to reference [37], the material coefficient in the formula $C_{10} = 0.11$, $C_{20} = 0.02$.

*3.2. Establishment of the Mechanical Model of Hexagonal Prism Soft-Bodied Bionic Actuator*

The hexagonal prism soft-bodied bionic actuator can realize axial bending deformation and radial expansion deformation respectively by using different inflation patterns. This section only establishes the mathematical model of the actuator bending deformation,

and the radial expansion deformation of the actuator is only verified by simulation and experiment later in the paper. By simplifying the model, the overall bending angle of the actuator can be regarded as the result of the superposition of the bending angles of each airbag in a single rectangular cavity. The equivalent diagram of actuator bending deformation is shown in Figure 4. Wherein, the overall bending angle of the actuator is represented by $\Phi$, the bending angle of a single airbag is represented by $\theta$, and the number of airbags is represented by $N$, then $\theta = \frac{\Phi}{N} = \frac{\Phi}{11}$.

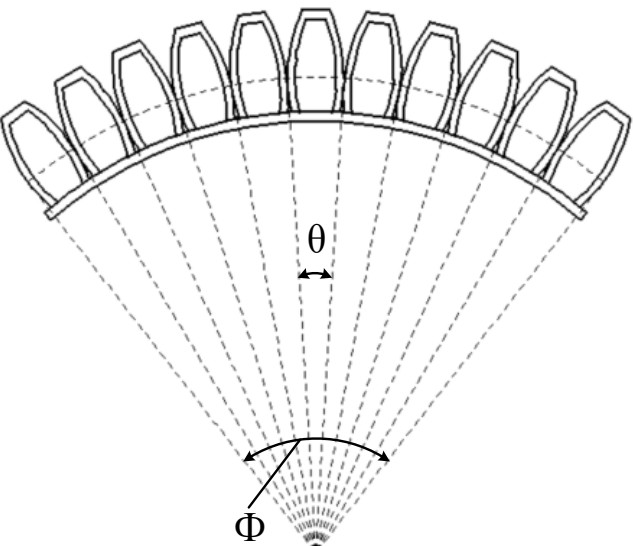

**Figure 4.** Equivalent model diagram of bending deformation of hexagonal prism soft-bodied bionic actuator.

The structural parameters of hexagonal prism soft-bodied bionic actuator are shown in Figure 5, where $l_1$ is the thickness of airbag expansion wall, $l_2$, $w_2$ and $h_2$ are the length, width, and height of airbag respectively, $h_1$ and $w_1$ are the thickness of airbag top wall and sidewall respectively, $h_4$ and $w_3$ are the width and height of cavity airway. $l$ is the length of a regular hexagonal prism, $a$ is side length, and $r$ is the radius of a hollow cylinder.

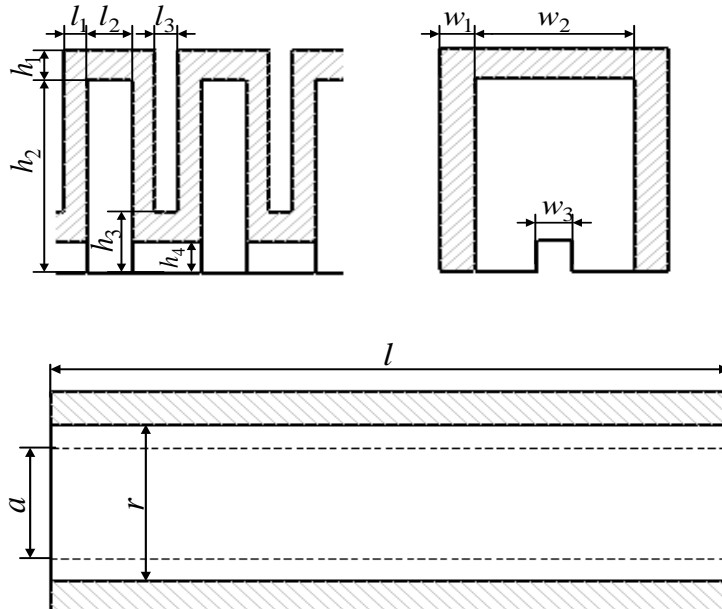

**Figure 5.** Schematic diagram of structural dimensions of hexagonal prism soft-bodied bionic actuator.

In order to simplify the analysis, the following assumptions are put forward for the modeling process of hexagonal prism soft-bodied bionic actuator:

The total volume of silicone rubber does not change before and after deformation;

The mechanical effect of the rectangular cavity on the passive deformation side of the hexagonal prism soft-bodied bionic actuator on the overall deformation process is not considered;

The curvature of the hexagonal prism soft-bodied bionic actuator changes uniformly in the process of bending deformation.

Based on the above assumptions, when other external forces are ignored and the bending deformation of the hexagonal prism soft-bodied bionic actuator reaches a stable state, the work done by the driving air pressure is completely used to overcome the internal stress of the silicone rubber material. According to the principle of virtual work, the following balance equation can be established:

$$PdV_A = V_R dW \tag{5}$$

where $P$ is the driving air pressure, $V_A$ is the volume of the air chamber, and $V_R$ is the volume of silicone rubber material. Where $V_R$ can be calculated by the following formula:

$$V_R = 2V_a + V_b \tag{6}$$

where $V_a$ is the volume of silicone rubber material in a single rectangular cavity, and $V_b$ is the volume of a hollow hexagonal prism. From Figure 4:

$$
\begin{aligned}
V_a &= N(L_a W_a H_a - l_2 w_2 h_2) + l_3 h_3 W_a (N-1) \\
V_b &= \left( \frac{3\sqrt{3}}{2} a^2 - \pi r^2 \right) l
\end{aligned}
\tag{7}
$$

where:

$$
\begin{aligned}
L_a &= 2l_1 + l_2 \\
W_a &= 2w_1 + w_2 \\
H_a &= h_1 + h_2
\end{aligned}
\tag{8}
$$

The volume $V_0$ of a single rectangular cavity before deformation is:

$$V_0 = NL_a W_a H_a + l_3 h_3 w_3 (N-1) \tag{9}$$

The volume of the air chamber of the deformed rectangular cavity is $V_A = 2(V_1 - V_a)$, and $V_1$ is the volume of a single rectangular cavity after deformation, which can be approximately calculated as follows:

$$V_A = 2((1+\lambda_1)V_0 - V_a) \tag{10}$$

The main elongation ratio of the rectangular cavity is:

$$\lambda_1 = \frac{\Phi}{\sin \Phi} \tag{11}$$

Based on the above analysis process, the relationship between the bending angle $\Phi$ of the soft-bodied actuator and the driving air pressure $P$ is obtained as follows:

$$P = \frac{(\sin \Phi)^2 V_R}{2V_0 (\sin \Phi - \Phi \cos \Phi)} \frac{dW}{d\Phi} \tag{12}$$

Finally, the $P - \Phi$ curve of the theoretical deformation model of the soft-bodied actor is obtained, as shown in Figure 6. It can be seen that the input pressure of the actuator is approximately linear with the output angle.

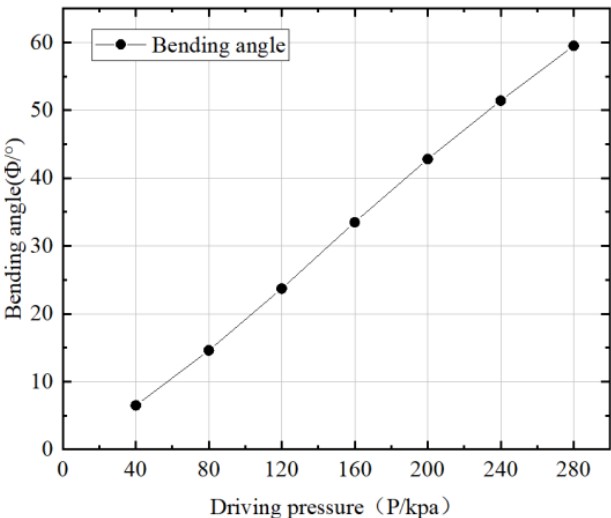

**Figure 6.** Bending angle–driving pressure curve of the hexagonal prism soft-bodied bionic actuator.

## 4. Verification of Numerical Simulation Algorithm and Preparation of Physical Model

*4.1. Numerical Simulation Algorithm Verification*

Considering the requirements of the motion mode of the peristaltic soft-bodied pipe robot on the deformation function of the actuator, the rationality of the structure and function of the hexagonal prism soft-bodied bionic actuator is verified by using the numerical simulation algorithm, and the accuracy of its deformation analysis theoretical model is tested [38–41].

Because the actuator is made of a hyperelastic material, its material characteristics have a nonlinear relationship. Compared with other simulation software, ABAQUS simulation software has great advantages in solving various nonlinear problems. Therefore, this simulation software is used to simulate and analyze the software driver. When ABAQUS is used, ABAQUS/standard is used for simulation analysis. The main steps in the simulation process are:

(1) Model establishment

The structure of each part of the soft actuator is established by using three-dimensional software, and imported into ABAQUS software to impose constraints for assembly.

(2) Soft-bodied actuator material and load setting

Since silicone rubber is a hyperelastic material, when defining the material characteristics, in addition to defining the density of the material, it is also necessary to set its hyperelasticity in the hyperelastics option, in which the strain energy function is Yeoh type, the silicone rubber model parameter is $C_{10} = 0.11$, $C_{20} = 0.02$, and the density is $\rho = 1130 \text{ kg/m}^3$.

When applying a load, first create a plane in the surface where the load is to be applied. Secondly, the load application is realized by creating the analysis step and setting the loads in the analysis step.

(3) Soft-bodied actuator meshing

When dividing the mesh, the method of dividing the model is adopted, and the hexahedron is used to sweep the mesh. When setting the mesh, because the silica gel is incompressible and has the characteristics of large deformation under stress, the mesh is set as a secondary hybrid solid element (C3D10H).

(4) Create a working file and conduct simulation analysis to obtain the deformation analysis diagram of hexagonal prism soft-bodied actuator, as shown in Figure 7.

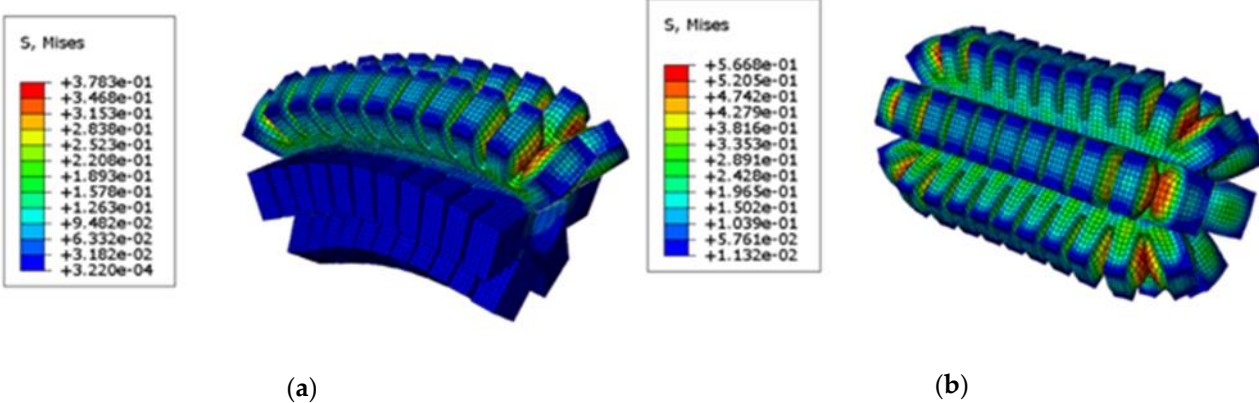

(**a**)                                                                 (**b**)

**Figure 7.** Deformation analysis diagram of hexagonal prism soft-bodied bionic actuator. (**a**) Bending deformation of hexagonal prism soft-bodied bionic actuator. (**b**) Expansion deformation of hexagonal prism soft-bodied bionic actuator.

According to Figure 7, the deformation of the hexagonal prism soft-bodied bionic actuator can meet the needs of the peristaltic soft-bodied pipe robot to realize multiple motion modes, and the deformation change law of the cloud diagram conforms to the actual deformation of the silicone rubber material after ventilation, which can verify the rationality of the structure and function of the actuator. Through numerical simulation calculation, the bending angle of the actuator under different driving pressures from 40 kPa to 280 kPa is obtained, as shown in Table 2.

**Table 2.** Numerical simulation algorithm calculation results.

| Driving pressure ($P/\mathrm{kpa}$) | 40 | 80 | 120 | 160 | 200 | 240 | 280 |
|---|---|---|---|---|---|---|---|
| Bending angle ($\theta/°$) | 5.5 | 13.7 | 22.6 | 32.5 | 41.5 | 47.4 | 50.5 |

The calculation results in Table 2 are compared with the theoretical model of driver bending deformation, as shown in Figure 8.

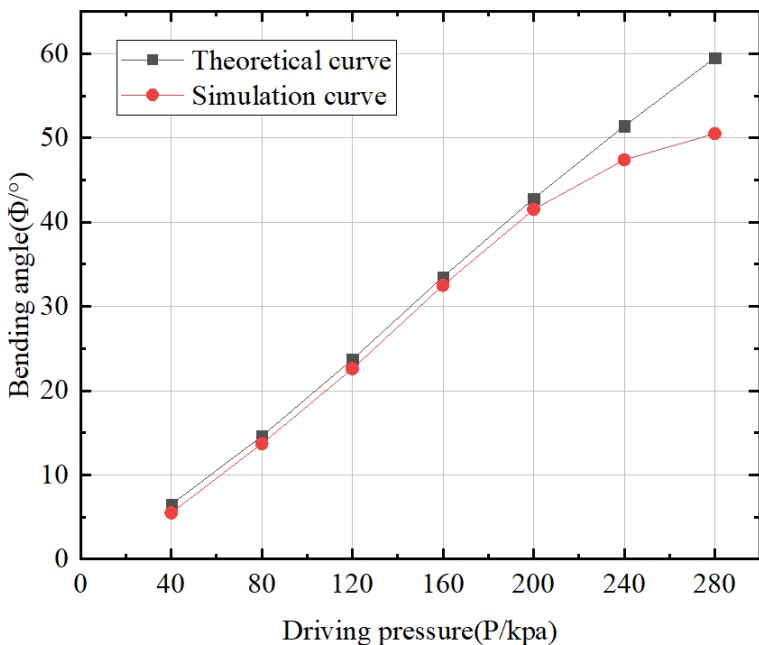

**Figure 8.** Comparison between simulation curve and the theoretical curves.

According to Figure 8, when the driving pressure is not greater than 200 kPa, the calculation result curve of the numerical simulation algorithm approximately coincides with the curve of the theoretical model, which can prove that the theoretical model of deformation analysis has a certain accuracy. The driving pressure continues to increase, and the error between the two curves becomes larger. After analysis, this was found to be because the influence of the rectangular cavity on the passive deformation side of the hexagonal prism soft-bodied bionic actuator on the bending deformation is not considered in the theoretical model analysis.

*4.2. Physical Model Preparation*

At present, there are two main manufacturing methods for pneumatic soft-bodied actuators: mold pouring type and 3D printing type [42]. Among them, 3D printing silicone products can only produce products with a relatively simple structure, and the cost is high. Therefore, this paper uses the mold pouring method to make the hexagonal prism soft-bodied bionic actuator. The materials required for the preparation method are shown in Table 3.

**Table 3.** Preparation of materials.

| Preparation of Required Consumables | Material Details |
| :---: | :---: |
| Rectangular cavity | Shore A45 platinum silicone rubber |
| Hexagonal silicone rubber column | Shore A45 platinum silicone rubber |
| Pouring mould | PLA |
| adhesive | silicone adhesive |

The physical model preparation process of the hexagonal prism soft-bodied bionic actuator is shown in Figure 9. Firstly, the cavity mold must be designed, and then printed by 3D printer; then, mix the platinum silicone rubber of components A and B to a ratio of 1:1, and the stock solution is obtained after uniform stirring; place the stock solution in a vacuum environment to remove bubbles, then pour the stock solution into the prepared mold, and remove bubbles for a second time. After removing bubbles, put the mold containing the stock solution into a 60 °C incubator and stand for 2 h. After curing and forming, demould to obtain the mold cavity; finally, the silicon rubber adhesive is used to bond each part of the cavity in order to obtain the hexagonal prism soft-bodied bionic actuator.

In view of the dual requirements of a pipeline and soft-bodied robot, it is necessary to ensure that the prepared actuators not only have flexibility, but also have a certain strength and wear resistance. Through a large number of data analyses and experimental tests, Shore A45 platinum silicone rubber was chosen as the preparation material for the hexagonal prism soft-bodied bionic actuator.

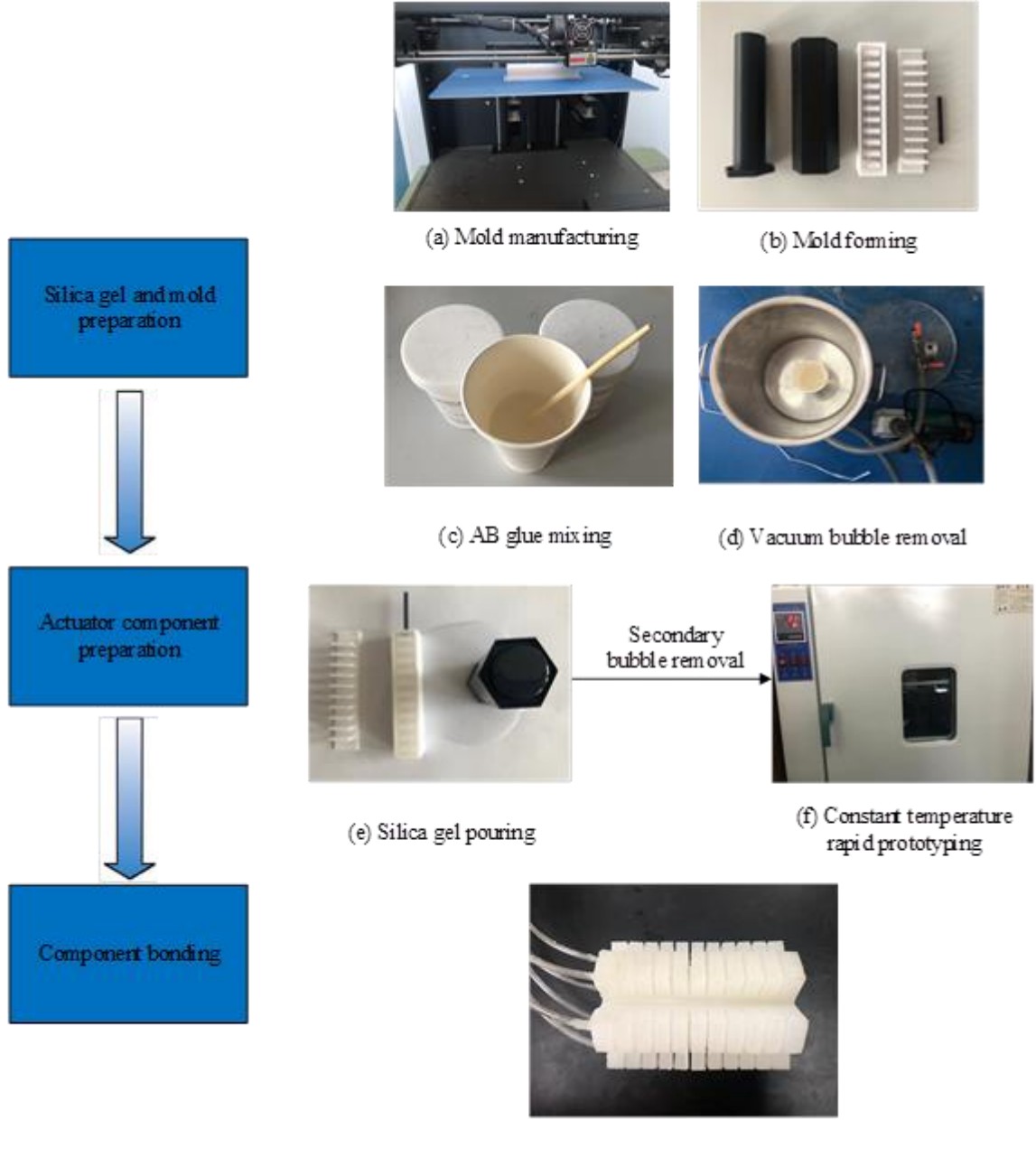

**Figure 9.** Physical preparation process of the hexagonal prism soft-bodied bionic actuator.

## 5. Experimental Test and Analysis of Hexagonal Prism Soft-Bodied Bionic Actuator

Through experimental test and analysis, we verified the different deformation effects of the hexagonal prism soft-bodied bionic actuator in the pipeline, and explored the influence of driving pressure on the motion and dynamic characteristics of the actuator, which can lay a foundation for the further optimal application of the actuator. The experimental device is shown in Figure 10. It mainly includes a hexagonal prism soft-bodied bionic actuator, air pump, manual pressure regulating valve, angle ruler, electronic dynamometer, fixture, and acrylic tubes with different pipe diameters.

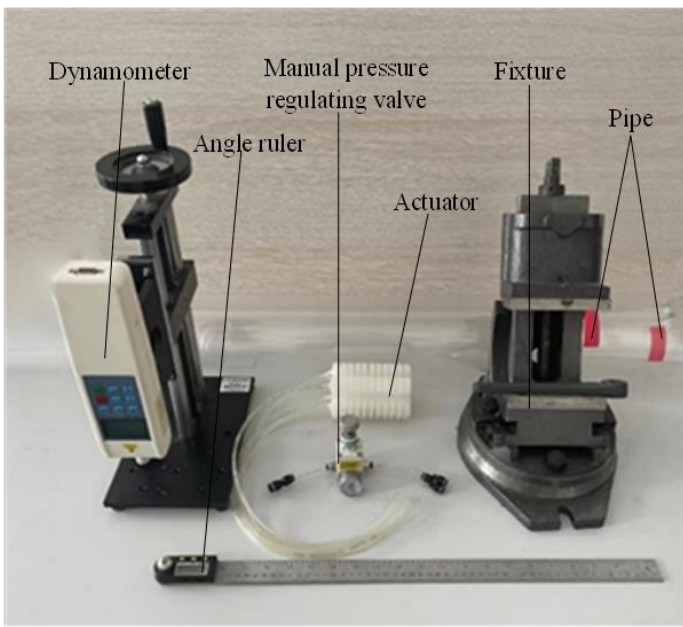

**Figure 10.** Experimental device diagram.

Different inflation strategies were adopted for the hexagonal prism soft-bodied bionic actuator in a pipeline with inner diameter of 84 mm and 115 mm respectively, and the deformation diagram of the actuator as shown in Figure 11 was obtained.

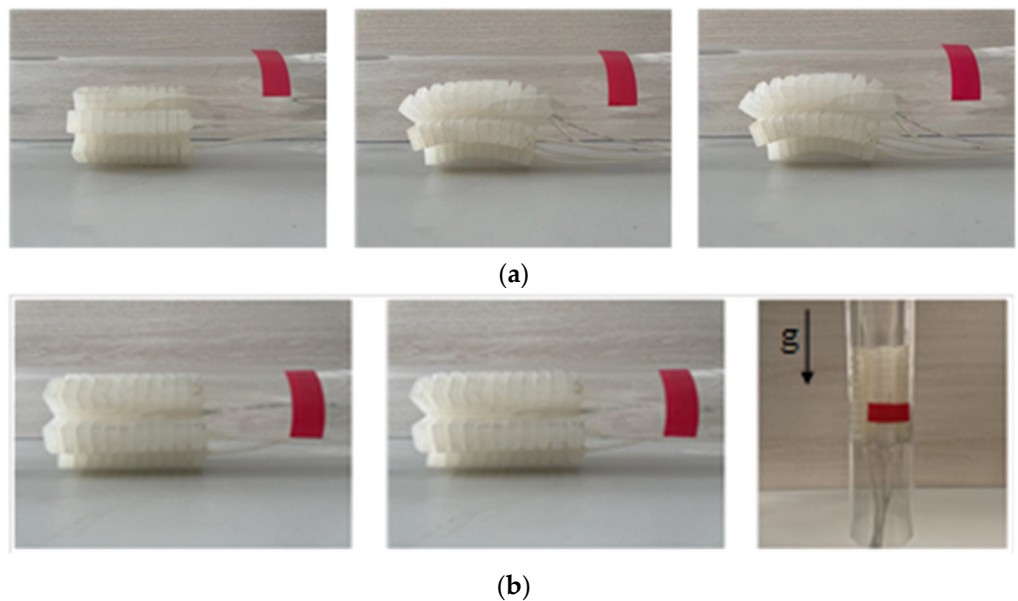

**Figure 11.** Pipe deformation diagram of hexagonal prism soft-bodied bionic actuator. (**a**) The actuator bends and deforms in the pipe. (**b**) The actuator expands and deforms in the pipe.

It can be seen from Figure 11 that the deformation mode of the hexagonal prism soft-bodied bionic actuator can meet the functional requirements of the multi-motion mode soft-bodied pipe robot, and the inflation strategy of double cavity inflation is adopted to give the actuator stability in the pipe.

*5.1. Experimental Analysis of Axial Bending Deformation of Hexagonal Prism Soft-Bodied Bionic Actuator*

5.1.1. Analysis of Motion Characteristics of Hexagonal Prism Soft-Bodied Bionic Actuator

The soft-bodied actuator is preset with different driving pressures through the pressure regulating valve, so that the hexagonal prism soft-bodied bionic actuator has different degrees of bending deformation, and then the digital display angle ruler is used to collect the bending angle data. Finally, the $P - \Phi$ characteristic curve of the driving pressure and bending angle of the actuator can be obtained, as shown in Figure 12.

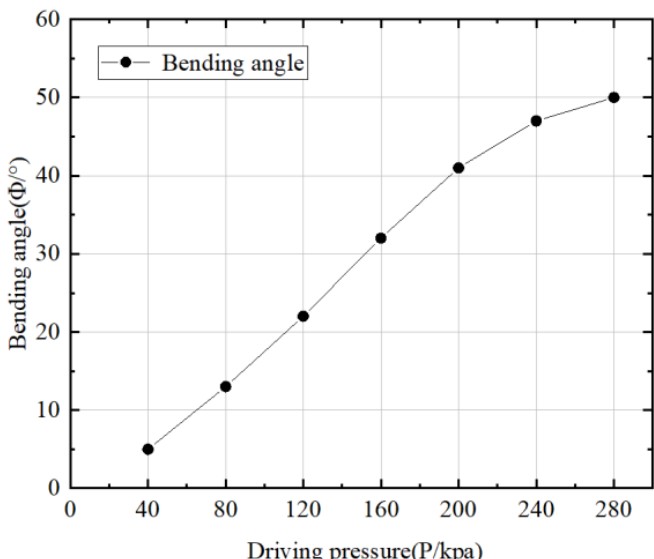

**Figure 12.** The characteristic curve of the hexagonal prism soft-bodied bionic actuator.

It can be seen from Figure 12 that the bending angle range of the hexagonal prism soft-bodied bionic actuator is $0 - 50°$. Among them, the change of $P - \Phi$ characteristic curve shows a certain linear relationship, but when the driving air pressure exceeds 200 kPa, it shows a non-linear change trend, because when the driving air pressure reaches 200 kPa, each air cavity of the rectangular cavity on the bent side begins to contact. When the pressure continues to increase, the interaction between the air cavities increases the counter bending moment of the actuator, hindering the effect of bending deformation. This can also explain the reason for the error between the simulation results and the theoretical model when the driving pressure is greater than 200 kPa.

According to the $P - \Phi$ characteristic curve, the empirical formula reflecting the characteristic change can be obtained, and its expression is:

$$f(x) = p_1 \cdot x^4 + p_2 \cdot x^3 + p_3 \cdot x^2 + p_4 \cdot x + p_5 \tag{13}$$

The specific coefficients in the formula are shown in Table 4.

**Table 4.** Empirical formula coefficient of the characteristic curve.

| Coefficient | $p_1$ | $p_2$ | $p_3$ | $p_4$ | $p_5$ |
|---|---|---|---|---|---|
| | $1.1097 \times 10^{-8}$ | $-9.4894 \times 10^{-6}$ | 0.0024 | 0.0146 | 1.2143 |

Error analysis was performed on the characteristic curve of $P - \Phi$ obtained from the experimental test, the calculation result of the numerical simulation algorithm and the theoretical model, as shown in Figure 13.

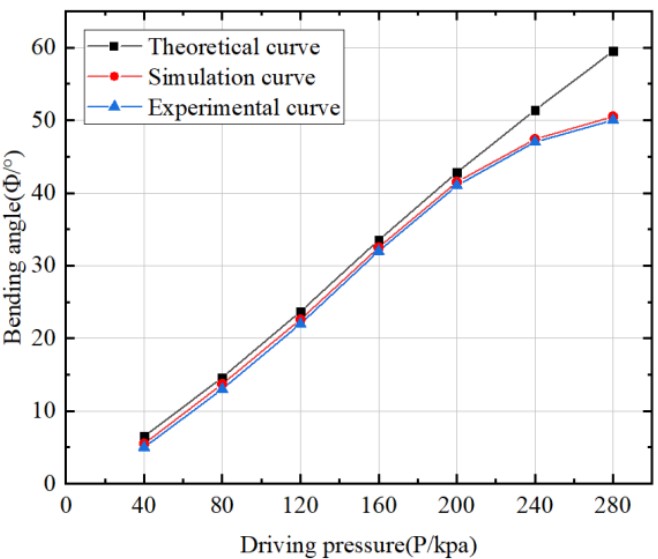

**Figure 13.** Actuator bending deformation error analysis diagram.

It can be seen from Figure 13 that the coincidence degree between the simulation curve and the experimental curve is high, which can prove that the numerical simulation calculation method is correct, and the theoretical model of deformation analysis is also accurate when the driving pressure is not greater than 200 kPa.

5.1.2. Dynamic Characteristics Analysis of Hexagonal Prism Soft-Bodied Bionic Actuator

The driving torque generated by the bending deformation of the hexagonal prism soft-bodied bionic actuator is collected by the electronic dynamometer, as shown in Figure 14.

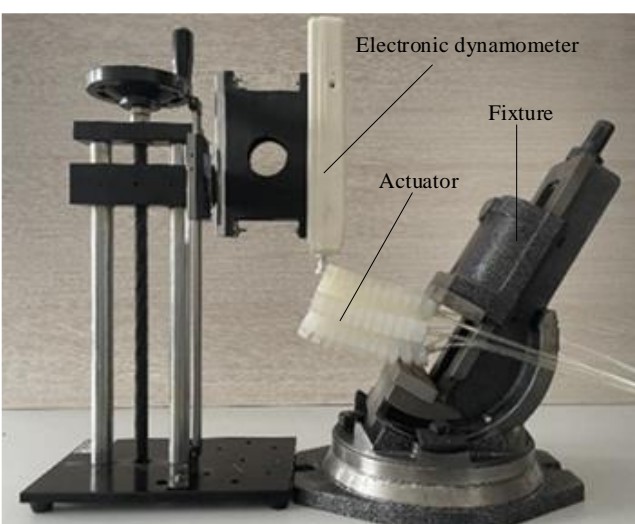

**Figure 14.** Driving moment measurement of hexagonal prism soft-bodied bionic actuator.

The $P - M$ characteristic curve of driving pressure and driving moment is shown in Figure 15. It can be seen from the figure that with the increase in driving pressure, the output moment of the actuator also increases. Conversely, when the rectangular cavity on the bent side begins to contact, the increase range of output moment begins to decrease with the increase in driving pressure.

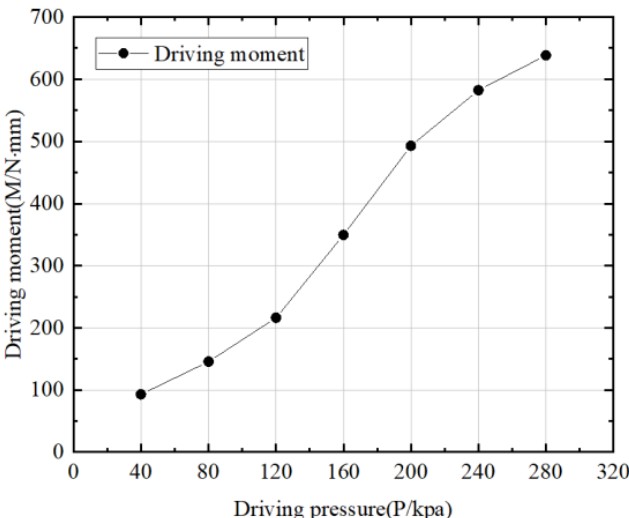

**Figure 15.** A characteristic curve of the hexagonal prism soft-bodied bionic actuator.

For the designed soft-bodied pipe robot, the larger the bending angle and output moment of the hexagonal prism soft-bodied bionic actuator the worse, as the contact area between the actuator and the pipe decreases with the increase in bending angle. However, in the experiment, it is found that when the driving pressure is 80 kPa, the contact area between the actuator and the pipeline reaches the minimum value. With the increase in the driving pressure, the contact area does not change, but the output moment will continue to increase, and then the friction between the actuator and the pipeline will increase, and the high driving pressure will reduce the service life of the actuator and increase the maintenance cost. Therefore, 80 kPa can be set as the driving pressure of the axial bending deformation mode of the hexagonal prism soft-bodied bionic actuator.

The $P − M$ characteristic curve is expressed by empirical Formula (13), and its specific coefficients are shown in Table 5.

**Table 5.** Empirical formula coefficient of $P − M$ a characteristic curve.

| Coefficient | $p_1$ | $p_2$ | $p_3$ | $p_4$ | $p_5$ |
|---|---|---|---|---|---|
| | $-1.4254 \times 10^{-7}$ | $1.8295 \times 10^{-5}$ | 0.0163 | $-0.9734$ | 107.200 |

*5.2. Experimental Analysis of Radial Expansion Deformation of Hexagonal Prism Soft-Bodied Bionic Actuator*

Taking the maximum circumscribed circle diameter of the hexagonal prism soft-bodied bionic actuator as the measurement size, the radial expansion deformation of the hexagonal prism soft-bodied bionic actuator occurs to varying degrees by changing the size of the air pressure introduced into the hexagonal prism soft-bodied bionic actuator, and the data is collected to obtain the characteristic curves of the driving pressure and the expansion dimension $P − D$, as shown in Figure 16.

It is known that the diameter of the selected experimental pipe is 84 mm. It can be seen from Figure 16 that when the driving pressure is 150 kPa, the diameter of the actuator becomes 84 mm. When the pressure continues to increase, the actuator will be fixed in the pipe. The greater the driving pressure, the better the fixing effect, but too much driving pressure is not the better, as overly high pressure will cause irreversible damage to the actuator. According to the motion requirements of the peristaltic soft-bodied pipe crawling robot, the minimum driving pressure of the drive at different inclination angles of the pipe is obtained through experiments, as shown in Table 6. With the increase in inclination angle, the driving pressure required to fix the actuator is greater.

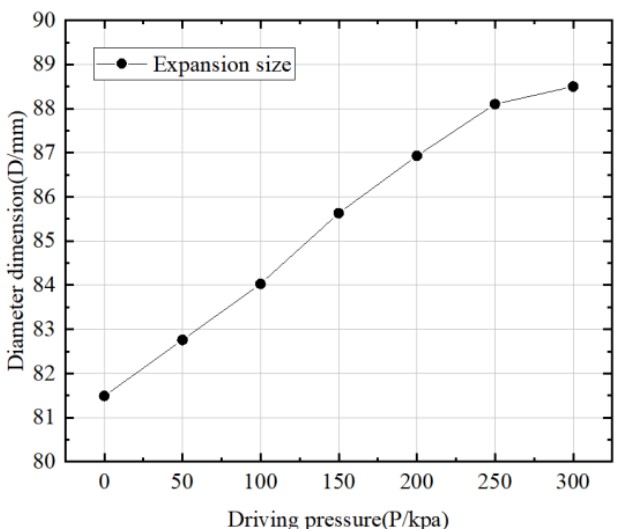

**Figure 16.** A characteristic curve of the hexagonal prism soft-bodied bionic actuator.

**Table 6.** Comparison table of pipe inclination and minimum driving pressure.

| Tilt Angle (°) | 0 | 30 | 60 | 90 |
|---|---|---|---|---|
| Driving pressure (kPa) | 160 | 175 | 195 | 225 |

## 6. Conclusions

According to the requirements of multi-motion modes for peristaltic prism soft-bodied pipe robots, a hexagonal prism soft-bodied bionic actuator is designed. By using different inflation strategies, the actuator can achieve different deformations.

(1) Based on the Yeoh binomial parameter silicone constitutive model, a deformation analysis model of the actuator is established to obtain the relationship between the bending angle of the actuator and the driving pressure, which can provide a theoretical reference for the structural design and deformation analysis of the soft-bodied actuator.

(2) Using numerical simulation technology, the rationality of the structure and function of the actuator is verified. When the driving pressure is not more than 200 kPa, the comparison error between the numerical simulation results and the theoretical model is 7.18%, which verifies that the deformation analysis model has a certain accuracy.

(3) An experimental platform is built to verify and analyze the performance of the prepared actuator. According to the input driving air pressure value, matching a certain range of motion and power output, the characteristic curves of actuator bending angle and driving moment are obtained, and the empirical formula is fitted. The accuracy of the deformation analysis model and numerical simulation algorithm is verified, and the radial expansion deformation characteristics of the actuator are experimentally studied. Finally, it is determined that the actuator is reasonable and feasible, and can be used as the driving mechanism of a multi-motion mode soft-bodied pipe crawling robot.

**Author Contributions:** Conceptualization, N.W. and Y.Z.; methodology, N.W. and G.Z.; software, N.W. and L.P.; validation, N.W. and W.Z.; formal analysis, N.W. and Y.Z.; investigation, N.W.; resources, Y.Z. and W.Z.; data curation, N.W. and L.P.; writing—original draft preparation, N.W.; visualization, N.W. and W.Z.; supervision, Y.Z.; project administration, N.W. and Y.Z.; funding acquisition, Y.Z.; All authors have read and agreed to the published version of the manuscript.

**Funding:** This work was supported by the National Natural Science Foundation of China (General Program) [grant number 52005344].

**Data Availability Statement:** All data generated or analyzed during this study are included in this article.

**Conflicts of Interest:** The authors declare that there is no conflict of interest regarding the publication of this paper.

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
