# Peer review of "Development and Analysis of Key Components of a Multi Motion Mode Soft-Bodied Pipe Robot"

_actuators, doi:10.3390/act11050125_

Round 1

Reviewer 1 Report

see attached manuscript with annotations.

Reviewer 2 Report

This paper is interesting and instructive to other researchers. Before publishing, I have some comments.

  1. The authors should emphasize the importance of using pneumatic systems. There are many actuating methods, like hydraulic or electro-hydraulic methods. They should mention these kinds of methods.

Kellaris, N., Gopaluni Venkata, V., Smith, G. M., Mitchell, S. K., & Keplinger, C. (2018). Peano-HASEL actuators: Muscle-mimetic, electrohydraulic transducers that linearly contract on activation. Science Robotics, 3(14), eaar3276.

Takeshi Iizuka, and Shingo Maeda. "Bidirectional electrohydrodynamic pump with high symmetrical performance and its application to a tube actuator." Sensors and Actuators A: Physical 332 (2021): 113168.

  1. 1 is not the structural design. Instead, it is principal design.

  1. For the motion, only one or two prisms are enough for your design. I do not know why you need six? and, in the demonstration, the authors only use two prisms for the motion.

  1. Chinese characters appear in Figure. 7. In the title, what is the meaning of soft-bodied-boided ?

  1. In fig. 8 and fig.13, what is the meaning of comparison between two theoretical parts?

  1. This paper should be arranged carefully again and English should be checked by native speakers. Otherwise, it is hard to understand.

Round 2

Reviewer 2 Report

The authors addressed my questions carefully.